# A National Innovation System Concept-Based Analysis of Autonomous Vehicles' Potential in Reaching Zero-Emission Fleets

Nalina Hamsaiyni Venkatesh and Laurencas Raslavičius *

Department of Transport Engineering, Faculty of Mechanical Engineering and Design, Kaunas University of Technology, LT-51424 Kaunas, Lithuania; nalina.venkatesh@ktu.lt
* Correspondence: laurencas.raslavicius@ktu.lt

**Abstract:** Change management for technology adoption in the transportation sector is often used to address long-term challenges characterized by complexity, uncertainty, and ambiguity. Especially when technology is still evolving, an analysis of these challenges can help explore different alternative future pathways. Therefore, the analysis of development trajectories, correlations between key system variables, and the rate of change within the entire road transportation system can guide action toward sustainability. By adopting the National Innovation System concept, we evaluated the possibilities of an autonomous vehicle option to reach a zero-emission fleet. A case-specific analysis was conducted to evaluate the industry capacities, performance of R&D organizations, main objectives of future market-oriented reforms in the power sector, policy implications, and other aspects to gain insightful perspectives. Environmental insights for transportation sector scenarios in 2021, 2030, and 2050 were explored and analyzed using the COPERT v5.5.1 software program. This study offers a new perspective for road transport decarbonization research and adds new insights to the obtained correlation between the NIS dynamics and achievement of sustainability goals. In 2050, it is expected to achieve 100% carbon neutrality in the PC segment and ~85% in the HDV segment. Finally, four broad conclusions emerged from this research as a consequence of the analysis.

**Keywords:** National Innovation System; autonomous vehicles; zero-emission fleet; energy sector; transport sector; industry transformation; COPERT software; decarbonization; transport roadmap





## 1. Introduction

The imminent challenges relevant to the implementation and utilization of autonomous mobility (AM) in daily life remain a global concern, raising interest from industrial and academic experts to address specific uncertainties. Autonomous vehicles (AVs) gained momentum in the research and adoption process, as they are advantageous over conventional vehicles in aspects of energy efficiency [1,2], reduction in emissions from driving toward a climate-neutral environment [3–5], and the declension of road fatalities [6–8]. The progress toward autonomous mobility in a global perspective is considered achievable by 2050. The evaluation of the geographical progress in terms of research and implementation exposes the challenges encountered by countries that demand case-specific analysis factors, such as National Innovation Strategy (NIS) [9–12]. To realize this approach, a level of the structure–object analysis "is split" into the number of sublevels (into a set of strata) [13]. The successful adoption of AVs is integral to parallel development of smart road infrastructures; this is what makes the application of the NIS approach very challenging and different compared to the same method applied to assess the electric vehicle penetration levels alone. As described in [14], National Innovation Systems are not, however, closed systems. They interplay with the driving forces of technological and policy development outside national boundaries [14,15], especially when related to a consumer product of a global industry [15]. However, the disposition of different indicators for a particular region, such

as the Readiness for Frontier Technologies, IMD World Digital Competitiveness Ranking, Government Artificial Intelligence (AI) Readiness Index, Autonomous Vehicle Readiness Index, and Global Innovation Index, coupled with insights from the roadmaps of the energy and transport sectors makes the prediction on autonomous transport development trajectories and their expected impact on decarbonization scenarios more reliable. To better understand the conditional relationship between the indicators listed above and the scale of innovations in the field of autonomous mobility, latest developments in the United States, the United Kingdom, European Union countries, Israel, the Republic of Korea, Singapore, and Japan are briefly reviewed.

The United States is leading in technological advancement and adoption that could potentially be the largest incubator for reforming technology of autonomous mobility. The Readiness for Frontier Technologies Index in the geographical aspect infers that the US is the pioneering country with a world ranking as the first with an index score of 0.94 in 2023 [16]. The country has also maintained the first position in the IMD World Digital Competitiveness Ranking since 2023 [17]. The official roadmap for planning autonomous mobility in the US stated that by the end of 2020, manufacturers will offer mature versions of self-driving cars for everyday use, and by 2040, 50% of the vehicle fleet will be made up of AV [18,19]. Perspectives in a larger context include the introduction of shared autonomous mobility, where one of these vehicles could replace 9–11 personally owned cars [16]. It is expected that a reduction in fleet size of 43% can be achieved with the adoption of autonomous mobility, while this change will affect the habits of parking, use, transit, charging, pedestrian behavior, and cycling on the road [18]. The states of California and Arizona have 67 AV test permit holders with 881 test vehicles that have 6.5 million trial miles, along with Waymo and Nuro companies with a permit to test without drivers, and Zoox, AutoX, Pony.ai, Waymo, Aurora Innovation, Cruise, and Voyage have permits to carry passengers [20].

The European countries strive toward a zero-carbon fleet to achieve climate neutrality by 2050, having defined energy reforms, policies, and measures, with a tendency to evolve as the world leader for autonomous mobility systems in a decade. It is expected that the passenger car (PC) fleet, which consisted of only 1.4 million clean vehicles in 2015, will increase to approximately 245–250 million clean vehicles (Scenario #1: ~170 mln. of Battery Electric Vehicle (BEV)/AV + ~75 mln. of fuel-cell vehicles; Scenario #2: ~225 mln. of BEV/AV + ~25 mln. of FCEV) by 2050 with a projected reduction in $CO_2$ emissions by 473 $MtCO_2$ (95%) in comparison to 2021 in the EU [21]. Report [22] shows that an investment of EUR 28 trillion in sustainable mobility is required to reach net zero emissions by 2050 in the 27 EU member states alone, which will yield a cumulative benefit of between EUR 340 and EUR 450 billion [21]. The collaboration between companies in the automotive industry in Sweden (Scania CV AB, AB Volvo, Volvo Car Group, and FKG), the Swedish Transport Administration, and the Swedish Energy Agency (jointly abbreviated as FFI) finances R&D activities for approximately SEK 0.1 billion per year. Several programs, namely 'Electronics, Software and Communication', 'Cyber-Security for Automotive', 'Electromobility', 'Road Safety and Automated Vehicles', and 'Efficient and Connected Transport System', are characterized as top priorities for the country to cope with the transformation in transportation [23]. The United Kingdom is one of the world leaders in the field of AM with a range including from industry to government investment in automated vehicle technology, the open regulatory regime, and the world-class research base. The UK is building resilient infrastructure and increasing access to transportation where the market for this sector can be worth EUR 48.9 billion by 2030, which will be 6% of the estimated EUR 757 billion global market prediction for this technology, creating 38 k new jobs [24]. Israel has introduced testing of AV as early as December 2017 when the Russian multinational technology company Yandex NV was approved to test. Volkswagen Group, Champion Motors, and Intel-owned Mobileye announced that they will use a global beta site 'New Mobility in Israel' for the provision of Mobility-as-a-Service (MaaS) once autonomous vehicles become available [20,25].

Significant global contribution in the field of research and testing of AV can be observed among Asian countries over the past decade. The Global Innovation Index for world economies in 2022 quantifies the Republic of Korea, Singapore, and Japan that demonstrated an outstanding collective impact on innovation capabilities, being in the 10th, 5th, 12th, and 13th place, respectively [26]. The steady position of fifth place held by Singapore can be considered the best country to adopt AV compared to the other three leading Asian countries. AVs were first on the policy agenda in 2013, stating that in the year 2022, three new towns (Punggol, Tengah, Jurong) will be the first to be equipped with driverless buses on the roads during off-peak hours [27]. Hyundai Motor Company reached level 4 AV in 2018 on a 190 km highway, followed by Kia Motors to develop a 5G-based vehicle-to-everything (V2X) (5G-V2X) [28]. South Korea rose six places from the previous year in 2022 and was ranked seventh in the Autonomous Vehicle Readiness Index, mainly due to the first commercialization of the 5G telecommunication service in the world [29,30]. The K-map scenario [31] reveals that the clean vehicle fleet in Korea will expand to a total of 10 million vehicles (5.3 M battery electric, 4 M plug-in hybrid, and 850 K hydrogen fuel cell) in 2030 and 24.3 million vehicles (18.8 M battery electric, 5.5 M clean vehicles) in 2050. This will drastically reduce road transport emissions by 29 $MtCO_2Eq$ in 2030 and by 78 $MtCO_2$ in 2050 [31]. The carbon price projections for Korea indicate that the benefits of reducing greenhouse gas emissions can range from EUR 0.1 trillion to EUR 0.2 trillion [29]. The transport sector in Japan has the second-best potential to reduce GHG among all industry sectors ($-100$ $MtCO_2Eq$ or $-48\%$ by 2030 compared to 2018 before reaching 100% $CO_2$ neutrality in 2050) [32]. A drastic increase in the number of electric vehicles and AV is estimated to be from approximately 1 million in 2020 to 12 million in 2030, and then 44 million in 2050 [32]. Report [32] states that the Japanese energy system has enough energy sources to meet these challenges.

Collectively, the leading countries in the field of the adoption of autonomous mobility exhibited a pattern of maintaining a significant position in the global ranking assessments of innovation, research, artificial intelligence and autonomous mobility readiness, energy targets, and decarbonization goals. Assuming that the concentration of innovation processes takes advanced positions in the structure of innovative entrepreneur activity [13], the dynamics of parameters of technological development of Lithuania were estimated in relation to the technological impact AVs will have on the decarbonization of the road transport in a long-term perspective.

Region-specific indicators, which characterize the NIS, such as total R&D spending, industry capacity, and major artificial intelligence reforms, together with policy documents and business interest, in Lithuania have been focused on and analyzed in depth. This study will exclusively provide readers, stakeholders, and investors with the opportunity to identify the untapped potential in Lithuania to be able to promptly foster and adopt frontier technology in the Baltic region, by addressing the following research questions:

(1) How can innovation activities in Lithuania in the field of autonomous mobility be analyzed in a broader National Innovation System?
(2) Do the Lithuanian energy targets suffice the need to address the wider adoption and implementation of autonomous vehicles?
(3) What are the expected transport fleet shift scenarios until 2050 in Lithuania and their effect on GHG emissions?
(4) What is the current standing of Lithuania among EU countries and what can be expected on a long-term horizon in view of the NIS analysis?

Collectively, this article will draw the interest of researchers, industry CXOs, and policy makers to understand the nuances that arise while moving down to the region level from a global perspective.

## 2. Materials and Methods

The country-specific data were collected from both primary and secondary sources. An extensive review of R&D projects, scientific publications, Lithuania's energy policy

documents, and relevant books, journals, reports, and case studies from LT and elsewhere in the developing world provided the theoretical basis for designing the field instruments and analysis of the data. The chosen impacting aspects that directly or indirectly significantly affect the adoption of AV and smart road infrastructure are examined in each section to find specific answers.

The NIS (step #1 in Figure 1) is presented as three interrelated macro blocs: the business environment, environment producing knowledge, and knowledge transfer mechanism. Based on the obtained findings and insights, we then calculate emission scenarios for passenger cars and light-duty and heavy-duty vehicles within three different time horizons: 2022, 2030, and 2050 (step #2). Compared to other case studies on decarbonization roadmaps, to give as much of a realistic picture for 2030 and 2050 scenarios as possible in terms of AV and smart infrastructure readiness, we employ the outcomes of the NIS analysis. Further on, Lithuania's standings in the context of EU countries are briefly assessed (step #3).

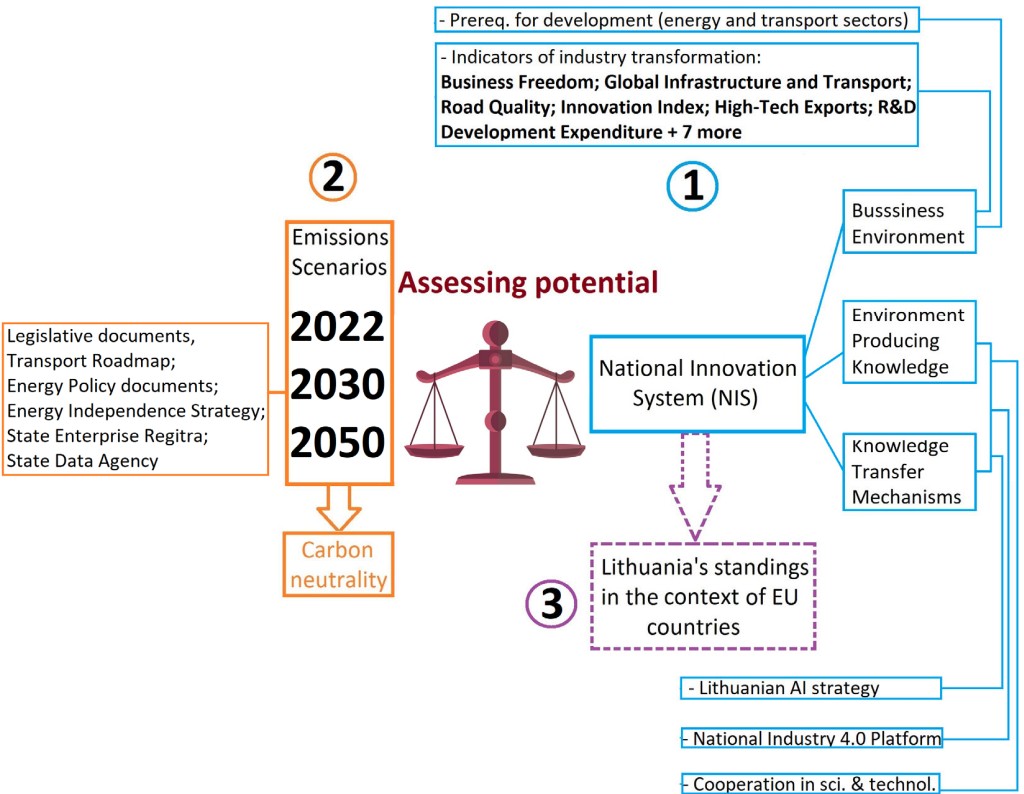

**Figure 1.** Schematic diagram of research design.

Emission estimations from road transportation (step #2 in Figure 1) were assessed based on the 2006 IPCC Guidelines Tier 3 method for $CO_2$ $CH_4$ and $N_2O$ calculations proposed in COPERT v5.5.1 software [33]. The main vehicle categories were allocated to the UNECE classification as follows:

- Passenger Cars M1.
- Light-Duty Vehicles N1.
- Heavy-Duty Vehicles N2, N3.

The Tier 3 method calculates emissions using a combination of firm technical data and activity data. The activity data of road transport were split and filled in for a range of parameters including the following [33]:

- Fuel consumed and quality of each fuel type.
- Emission controls fitted to a vehicle in the fleet.
- Operating characteristics (e.g., average speed per vehicle type and per road).

- Types of roads.
- Maintenance.
- Fleet age distribution.
- Distance driven (mean trip distance).
- Climate.

The statistical data show that approximately 440,316 personal cars belong to the countryside (60% of which participated in daily traffic) and 912.708 to a city (50% of which participated in daily traffic) in Lithuania [33]. It was assumed that rural/city traffic had specific contributions to different driving conditions: urban peak—15%/26%, urban off-peak—10%/26%, rural—50%/24%, and highway—25%/24%, respectively. The percentage of vehicles in everyday traffic was accounted for as 75% of all vehicles in each category (N1, N2, and N3) [34,35]. Specific contributions to different driving conditions for HDVs are as follows: urban peak—3% (3% for LDV), urban off-peak—20–20% (50% for LDV), rural—20–47% (47% for LDV), and highway—30–67% (0% for LDV). Therefore, as far as driving conditions are concerned, total emissions were calculated by means of the following equation [33]:

$$E_{TOTAL} = E_{URBAN} + E_{RURAL} + E_{HIGHWAY} \tag{1}$$

where $E_{URBAN}$, $E_{RURAL}$, and $E_{HIGHWAY}$ are the total emissions of any pollutant for the respective driving situations.

Emissions of $CO_2$ were calculated on a basis of the amount and type of fuel sold and its carbon content:

$$Emission = \sum[Fuel_a \cdot EF_a] \tag{2}$$

in which *Emission* denotes the emission of $CO_2$ (kg), $Fuel_a$ is fuel sold (TJ), $EF_a$ is the emission factor (kg/TJ), a is the type of fuel (natural gas, petrol, diesel, etc.)

The emission equation for $CH_4$ and $N_2O$ for Tier 3 is as follows (emission factor assumes full oxidation of the fuel):

$$Emission = \sum_{a,b,c,d}[Distance_{a,b,c,d} \cdot EF_{a,b,c,d}] + \sum_{a,b,c,d} C_{a,b,c,d} \tag{3}$$

in which *Emission* denotes the emission of $CH_4$ or $N_2O$, $EF_{a,b,c,d}$ is the emission factor (kg/km), $Distance_{a,b,c,d}$ is the distance travelled during the thermally stabilized engine operation mode (km), $C_{a,b,c,d}$ are the cold-start emissions (kg), *a,b,c,d* are the vehicle types.

Environmental insights for the scenarios in 2021 (current), 2030 (middle-term), and 2050 (long-term) were obtained and analyzed for two broad categories: (i) type of fuel and (ii) driving under specific conditions. Each scenario represents changes in vehicle fleet composition over the time horizon, demographic changes across the country, economic trends, and the alteration of specific habits and behaviors. In the 2030 scenario [34], it is expected that there would be a significant increase in the number of electric cars, which will constitute about 10% of the fleet. Document [34] suggests that in 2050, the majority of the fleet will comprise autonomous cars and trailers. Changes in fleet composition will definitely influence GHG trends, especially in a long-term perspective. The GHG emission from passenger cars (PCs) as well as light- and heavy-duty vehicles was evaluated for four different driving conditions (urban peak, urban off-peak, rural, and highway) and, depending on the vehicle category, it was evaluated for 2–5 types of fuel (diesel, petrol, petrol HEV, petrol PHEV, LPG). Petrol and diesel fuel can be referred to as the reference fuels, as they are dominant, whereas petrol hybrid, PHEV, biogas, and biodiesel are referred to as alternate fuels.

The uncertainty of the emission estimate is approximately ±10%, based on studies with reliable fuel statistics. The primary source of uncertainty is the activity data rather than emission factors.

## 3. Results

Following the chosen approach (step #1 in Figure 1), the National Innovation System is described and presented first in three interrelated macro blocs: the business environment, environment producing knowledge, and knowledge transfer mechanism. The first block, the business environment, is subdivided into two system structure–objects and their functional approaches to NIS performance (Sections 3.1.1 and 3.1.2), while the blocks "Environment producing knowledge" and "Knowledge transfer mechanism" have three common constituents (Sections 3.1.3–3.1.5). Further, in Section 3.2, the vision of Lithuania toward the decarbonization of the transport sector by 2050 is briefly discussed. Finally, Section 3.3 is dedicated to the analysis of Lithuania's standings in the context of EU countries.

### 3.1. The National Innovation System

#### 3.1.1. Assessment of Prerequisites for Further Development of Lithuanian Energy and Transport Sectors

*Energy efficiency targets for the transport sector.* The objective of achieving 50–55% of emissions in 2030 compared to 1990 was stated in the European Green Deal objectives [35]. This strategy plays a key role in contributing to the transition to the consumption of clean energy and the efficient use of resources. This policy globally aims to restrict carbon pricing and the risk of carbon leakage to achieve the targets. The wider deployment of efficient and electric vehicles, coupled with public fleet renewal, the adoption of sustainable mobility plans, and rail electrification, will gradually reduce the fossil fuel use and remain as the main funding targets in Lithuania [36], as indicated in Table 1.

**Table 1.** Energy efficiency targets for the Lithuanian transport sector until 2030 [36].

| Target Group | Description | Predicted Savings by 2030 |
|---|---|---|
| Sustainable urban mobility plans | Targets: Reduce car usage. Measures: Promote cycling, walking, public transport, and the use of alternative fuels. | 2.95 TWh |
| Fossil fuel | Targets: Increase fuel cost—petrol (+14.7%), diesel (+5.2%), LPG (+64.7%). Measures: Inflation in applicable excise duties and taxes on fuel consumption beyond EU norms. | 6 TWh |
| Electrification of rails | Targets: New electric power lines for 814 km of rail by 2030. Measures: EU funding during 2025–2035. | 3.36 TWh |
| Electric vehicles | Targets: 10% of registered and re-registered passenger cars by 2030 and 50% by 2050. Measures: EUR 4K subsidy for acquisition of new electric cars, and EUR 2K for used (max. 5 years) electric cars. | 6 TWh |
| Urban and suburban public fleet renewal | Targets: Implementing 150 electric-powered city and shuttle buses. Measures: EU funding during 2030–2025. | 0.393 TWh |
| Public fleet renewal and green procurement of clean vehicles | Targets: 60% of public fleet (M1, M2, and N1) and 80% of buses (M3) must be 'clean' by 2025; 100% of the fleet (M1, M2, and M3), 16% of heavy-duty vehicles (N2 and N3), and 50% of buses must be 'clean' by 2030. Measures: Public fleet switch. | 0.521 TWh |
| Efficient vehicles | Targets: 42% increase in the number of energy-efficient vehicles until 2030. Measures: EUR 1K subsidy. | 0.9 TWh |

The efforts to improve investment attractiveness in the energy and transport sectors and the encouragement of the participation of the largest private and public companies with a diversified portfolio of low-carbon businesses (low-carbon, green technologies) [37–41] in joint partnerships, coupled with other existing measures, such as Cohesion Policy funds [42], EU-ETS [43], and the Just Transition Fund [44] (First-of-A-Kind plant investments), can also be successfully employed [45].

*Projected energy demand*. The projected electricity demand forecast for Lithuania until 2050 is presented in Table 2 and Figure 2.

**Table 2.** Projected electricity demand in Lithuania [46].

| TWh | 2025 | 2030 | 2040 | 2050 |
|---|---|---|---|---|
| Annual gross electricity consumption | 13.64 | 14.47 | 17.77 | 19.81 |
| Annual gross load | 12.78 | 13.62 | 16.92 | 18.96 |
| Annual total load | 11.75 | 12.52 | 15.55 | 17.43 |

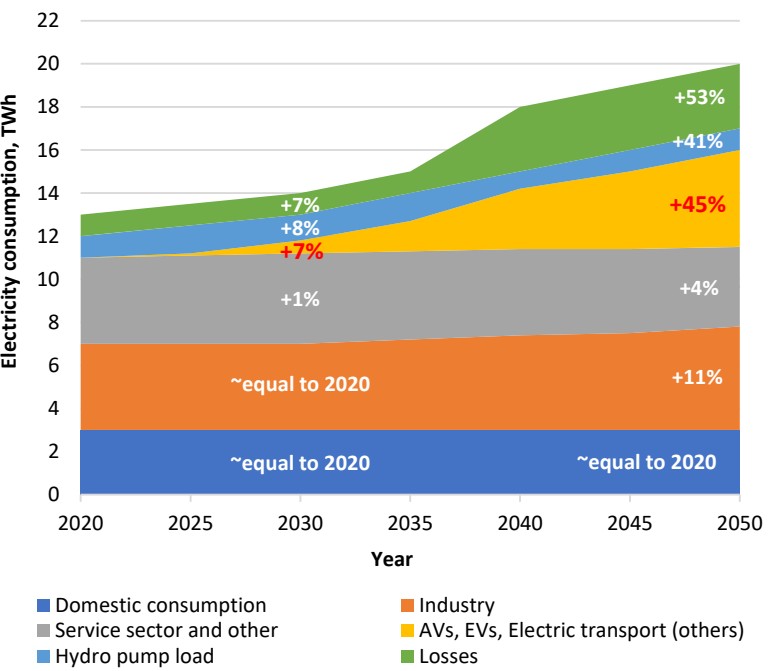

**Figure 2.** The roadmap for electricity demand in Lithuania for the period between 2020 and 2050 [46].

Electric energy consumption in the transport sector is expected to increase steadily until the year 2030, followed by a sharp jump by 27.6% between 2030 and 2040, and obtaining its peak of 19.81 TWh by 2050, which will exceed the 2020 levels by 45%. The second largest change in the pattern of electricity use during the 30-year period is expected to occur in the hydro pump load with a 41% higher request in 2050 than today. The short-term demand prediction for the year 2025 for the sectors remains closely equal to that of the figures in 2020; the upcoming years will face a 1% growth in industrial demand, followed by 7 to 8% higher consumption via the hydro pump load and transportation sector. There is a forecast that industry growth will experience a growth rate higher than the average until 2050 even though the expected demand for electricity in this category will increase by 11% [46]. The trend in electricity demand in Lithuania due to transport electrification between 2020 and 2050 is demonstrated in Figure 3a.

Lithuania had around 600 public electric charging stations or 0.2 EVC/100 km by the end of 2021, which is expected to increase by a factor of 10, accounting for a total of 6,000 charging stations or 2 EVC/100 km by the end of 2030. Lithuania is ranked among the EU countries with the least number of stations per 100 km of road, while The Netherlands, Luxembourg, and Germany are considered the leading countries in terms of EVC availability: 47.5 EVC/100 km, 34.5 EVC/100 km, and 19.4 EVC/100 km, respectively [47]. The increase in the access to charging for electric vehicles will allow the market share of electric vehicles to grow tremendously, coupled with the use of vehicle-to-grid technology in practice to be more efficient in using energy during periods of high customer demand [46], which can be seen in Figure 3b:

- 2020–2029: EVs have limited charging flexibility.

- 2030–2050: EV, mobility-as-a-service, and AV can change charging hours throughout the day.

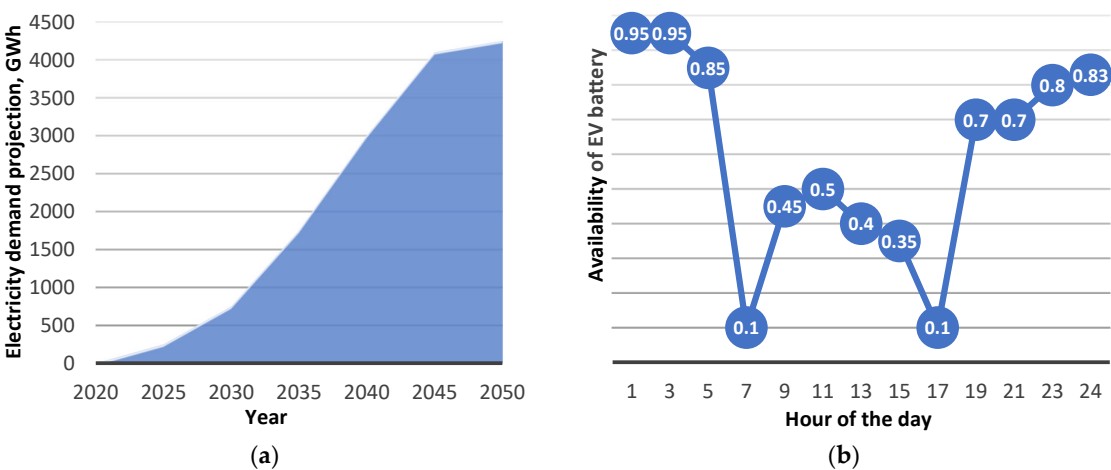

(**a**)  (**b**)

**Figure 3.** Evolution of electric mobility as a flexibility provider in Lithuania: (**a**) demand projection and (**b**) capacity profile (weekday) [46].

### 3.1.2. Indicators of Industry Transformation

The national competitiveness of a country is generally measured as an index that is evaluated on multiple phenomena that are diverse yet interrelated [48]. Similarly, to all developing countries, the cross-linkages between the transportation and manufacturing sectors, which are the crucial drivers of economic growth, depend on the scope of innovations, as new digital technologies and innovative solutions create huge opportunities [48,49]. This study attempted to assess the strongest positions of Lithuania according to different business and economic criteria for 200 world countries presented in [50,51]. Lithuania was on the 10th rank in 2020 among 175 counties (9th among EU countries) in both the Business Freedom category and the Economic Freedom, overall index [51]. Regarding the assessment of the global Infrastructure and Transport Characteristics, whose readiness level is of special importance for a wider deployment of autonomous vehicles, very strong positions were found in the following categories: the 20th rank among 138 countries in mobile network coverage (7th among EU countries) and Internet bandwidth (13th among EU countries), as well as the 38th rank in road quality (14th among EU countries) [51]. The country's efforts in the category Governance and Business Environment contributed to achieving one of the leading positions (sixth among EU countries) among 173 countries according to the Cost of Starting Business indicator. Lithuania ranked sixth among all EU countries with 0.5% of income per capita, overtaking countries such as France, Finland, Norway, Iceland, and others. Identified as one of the smallest countries in the world in terms of population (142nd) [52], Lithuania performs very well in the categories of High-Tech Exports (41st, 12.01% of manufactured exports or USD 2619.11 million in 2020), Information Technology Exports (27th, 3.92% of total goods exports), Research and Development Expenditure (35th, 0.94% of Gross Domestic Product (GDP)), and the Innovation Index (39th) [51]. The indices clearly represent national growth among global markets since 2007 and it is important to address the scope of innovation in economy building and technological advancements that could affect this index. The main aspects of the corresponding factors that assert the growth of the future index rankings in Lithuania would be the adoption of artificial intelligence (AI), the digitalization of industry, and the reform of mobility.

### 3.1.3. The Milestones of the Lithuanian Artificial Intelligence Strategy

Around 7 out of 10 companies (Small and Midsize Business and Enterprise) worldwide are predicted to adopt at least one type of AI technology by 2030, while almost half of corporate giants (more than 1,000 employees) would deploy the entire range [53]. The

broad classification for AI that will have been developing during the present decade includes the following distinctive categories: artificial narrow intelligence (all existing AI that has ever been created, which can recognize a voice but cannot drive a car) [54], reactive machines (do not have the ability to 'learn') [54], limited memory (all current assistants to self-driving vehicles use this technology) [55,56], theory of mind (the advanced level of AI systems) [56], self-aware (the final stage of AI development; AI superintelligence) [56]. The European Commission drafted and adopted a plan "made in Europe" [57] to promote the use of artificial intelligence systems in Europe in December 2018. The target sectors for the implementation of AI in Lithuania were indicated in [58] as follows: manufacturing, agriculture, healthcare, transportation, and energy.

The challenge of energy access can be broadly divided into three categories: energy production, predictive maintenance, and communication infrastructure [59,60]. AI techniques may outperform traditional models for energy production by defining local control strategies, automating the management of variable energy sources, and ensuring optimal continuous (incl. autonomous) energy generation [59]. Emerging technology (AI) could perform ongoing, autonomous, and continuous monitoring of local energy generation devices and energy distribution elements as predictive maintenance. The efficiency of communication infrastructure can be markedly improved if different variants of AI are installed to ensure optimal use of the wireless spectrum, reduce communication infrastructure, and better protect against active and passive cyber-attacks [59]. Regarding an efficient approach to the entire energy supply chain through the wider deployment of AI capabilities, the higher sustainability is seen through the reduced dependency on foreign sources of energy.

Different associations (The 5G Automotive Association), standardization organizations (The 3rd Generation Partnership Project, European Telecommunications Standards Institute (ETSI)—Intelligent Transport Systems (ITSs), ISO), and joint research projects (The 5G Infrastructure Public Private Partnership) are working on harmonizing driving automation, the electrification of vehicles, connectivity, and sharing economy with each other [61]. Transportation and logistics play a pivotal role in the Lithuanian economy. The widespread use of autonomous vehicles has the potential to significantly alter the logistical dynamics in the country. The expected breakthrough in the fields of cybersecurity, privacy, 5G, the Internet of Things, the data economy, the free flow of data [62], and AI systems can automate real-time traffic control and reduce travel times. The widespread use of autonomous vehicles, which are seen as a flagship use case for 5G deployment along European transport paths, has the potential to significantly alter the logistical dynamics in the country. The country is on the brink of a fortune reversal by being a part of the EU initiative '5G cross-border corridors for CAM' to test automated transport [62]. Lithuania is assigned to 2 of 12 European digital cross-border corridors, where connectivity and digital technologies can be tested and demonstrated [62]:

- EE-LV-LT Via Baltica (E67) Tallinn (EE)—Riga (LV)—Kaunas (LT)—Lithuanian/Polish border;
- LT-PL Via Baltica Kaunas-Warsaw (and further a national extension between Kaunas and Vilnius (LT)).

This will allow the nation to build experience in autonomous vehicles and prepare for risk mitigation and future policy changes. The breakthroughs in energy and transportation sectors and artificial intelligence will also significantly improve the overall efficiency of manufacturing companies [63].

### 3.1.4. National Industry 4.0 Platform

The wide-scale adoption of digitalization, 3D printing, cloud computing, the Internet of Things, big data, and edge computing to obtain a viable advantage over conventional industrial techniques and processes is termed Industry 4.0 [64]. Industry 4.0 ('Pramone 4.0' in Lithuanian) is the Lithuanian strategic step toward the implementation of the industry–government interaction for technology, infrastructure, and digitalization. The industrial shift will beneficially affect the carbon energy transition, increase job opportunities, broaden

innovation and its diffusion, broaden entrepreneurship, mobilize the private sector, and make growth inclusive. The entire ecosystem for industry digitalization and key findings of the Industry 4.0 benefits for Lithuania are summarized in Figure 4.

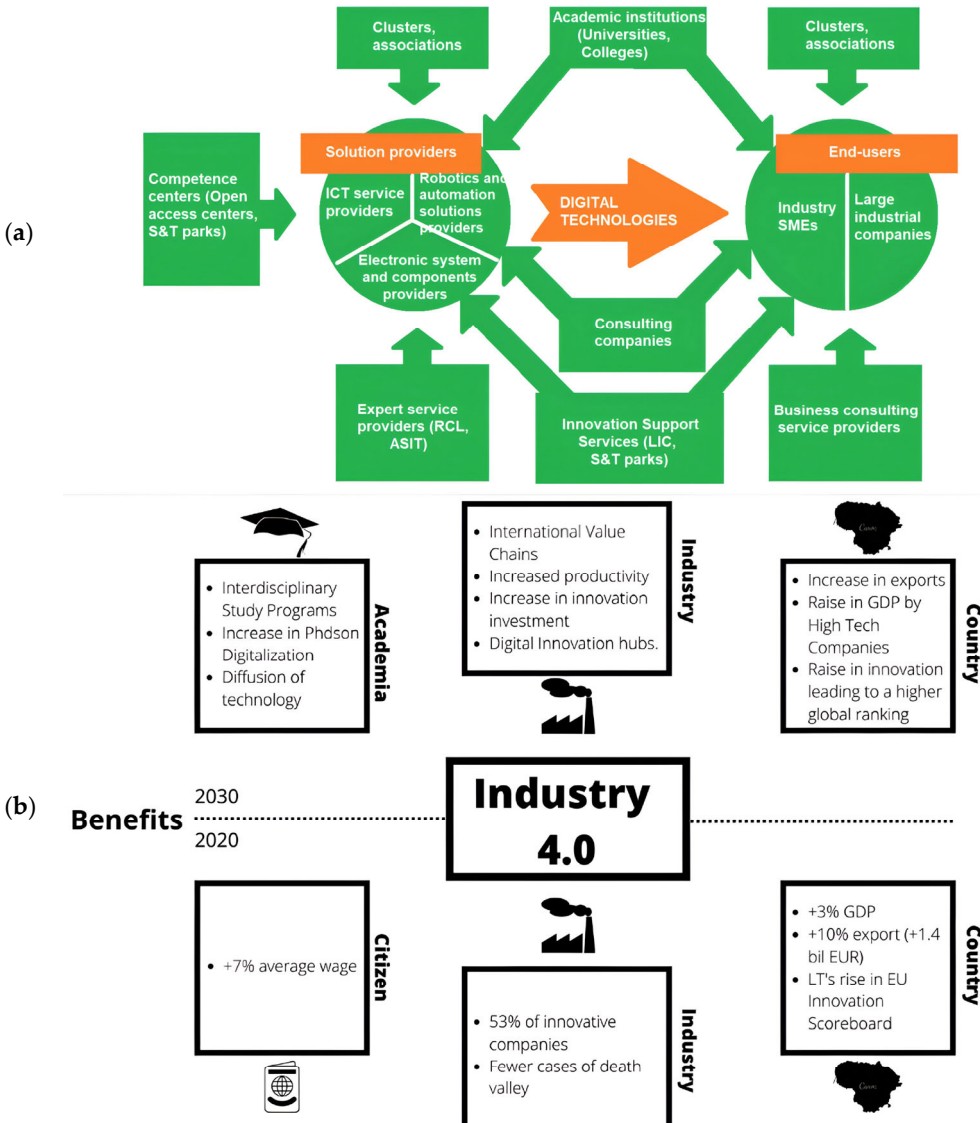

**Figure 4.** The roadmap for industry digitization 2020–2030: (**a**) existing ecosystem, (**b**) expected benefits.

The existing Lithuanian ecosystem connected to industry digitization initiatives in 2022 was based on three actors: (i) solution providers, (ii) end-users, and (iii) supporting organizations [65]. The national platform "Pramonė 4.0" was established [65] to ensure that the links between the different actors are effective in terms of investment and knowledge flows between all institutions. The Lithuanian digitalization vision for 2030 aims to focus on the dominant relatively small, smart, and agile factories that are capable of adapting flexibly to meet changing market needs while maintaining the production of high-value-added products. The national and international companies in the country are integrated with international value chains through ownership, production partners, and realization markets [66].

### 3.1.5. European Cooperation in Science and Technology: A Focus of Research for the Smart and Sustainable Mobility

Research in Lithuania contributes to the important aspects of traffic and transport planning with an inclusive approach that covers autonomous vehicles, intelligent transport systems, and cooperative intelligent transport systems to obtain the seamless functioning of mobility. The growing economy and new research opportunities are paving the way for the achievement of autonomous mobility on the roads soon. Globally challenging tasks cannot be solved solely by one country. In the European context, usually an approach stimulating the promotion of networking between experts (COST actions) in the field that aids to form a technically strong group for further research activities at-large are implemented under Horizon2020 (2014–2020) and Horizon Europe (2021–2027), which is tabulated in Table 3 [67]. While FP1-FP9 calls addressing the problems of sustainable mobility and electric vehicles as energy transition technology toward more environmentally friendly transportation systems, the H2020 initiative is more focused on the latest technological advances in the fields of connected and automated mobility (CCAM) and autonomous vehicles.

**Table 3.** The list of European Cooperation in Science and Technology projects on the topic of autonomous mobility and ITS [67].

| No. | Code | Title | Period | Involvement of Lithuanian Institutions |
|---|---|---|---|---|
| 1 | TU 1004 | Modelling public transport passenger flows in the era of intelligent transport systems | 2010–2015 | – |
| 2 | TU 1102 | Toward autonomic road transport support systems | 2011–2015 | – |
| 3 | TU 1302 | SaPPART satellite positioning performance assessment for road transport | 2013–2017 | – |
| 4 | CA 16222 | Wider impacts and scenario evaluation of autonomous and connected transport | 2017–2022 | KTU |

During the period 2015–2021, rapid progress in smart and sustainable mobility was achieved to align with Vision 2050—to reduce road deaths and GHG emissions by 80% and to decrease congestion levels and fossil fuel dependency to 40% by 2050. Vehicle positioning capabilities were evaluated during different road scenarios, including driving and platooning using onboard positioning systems [68] in 2015. The focus of the several projects [69–71] was to improve connected mobility systems by analyzing vehicle-to-infrastructure (V2I) systems for platooning, traffic management and vehicle assistance for smart mobility, and intrusion warning strategies for risk scenarios. The network-based development of employing 5G network V2X solutions to use a network slice, edge computing, semantic web technology, new radio features [72], and sensitive latency support [73] to bridge the feasibility gaps was among the main objectives of the research started in 2017–2018. The social response of AVs and the attributes associated with training and education and the characteristics of drivers are addressed in [74,75]. Three-level planning to achieve zero net emissions by 2050 [76,77], cyber security of CCAM [78], and the integration of driver skills for AVs [79] have revealed real-time challenges. Multimodal integration [80], the analysis of the AV business model [81], interoperability trials of CCAM [82], remote-assisted AVs for safety [83], and upgraded maps that include safety information for AVs [84] were analyzed in different projects in 2022. Autonomous airport logistics [85] and future multimodal traffic management scenarios for AVs [86] are the latest projects in progress toward inclusive transportation in the EU.

## 3.2. Emission Scenarios (The Vision of Lithuania toward Decarbonization of the Transport Sector by 2050)

Figure 5 shows the visualization of the annual GHG emissions in Lithuania that were equal to 2026.6 kton [87]. The transportation sector with 5996.5 kton of $CO_2$ emissions was the largest emitter among all industries, showing a 90% increase in the level since 2000. Two major emitters, passenger car transport (3360.8 kton) and heavy trucks and buses (2106.2 kton), were responsible for 81.3% of all $CO_2$ released into the atmosphere via the transportation sector. The contribution of an ever-increasing number of light-duty vehicles and heavy-duty vehicles (LDVs, HDVs) is paramount.

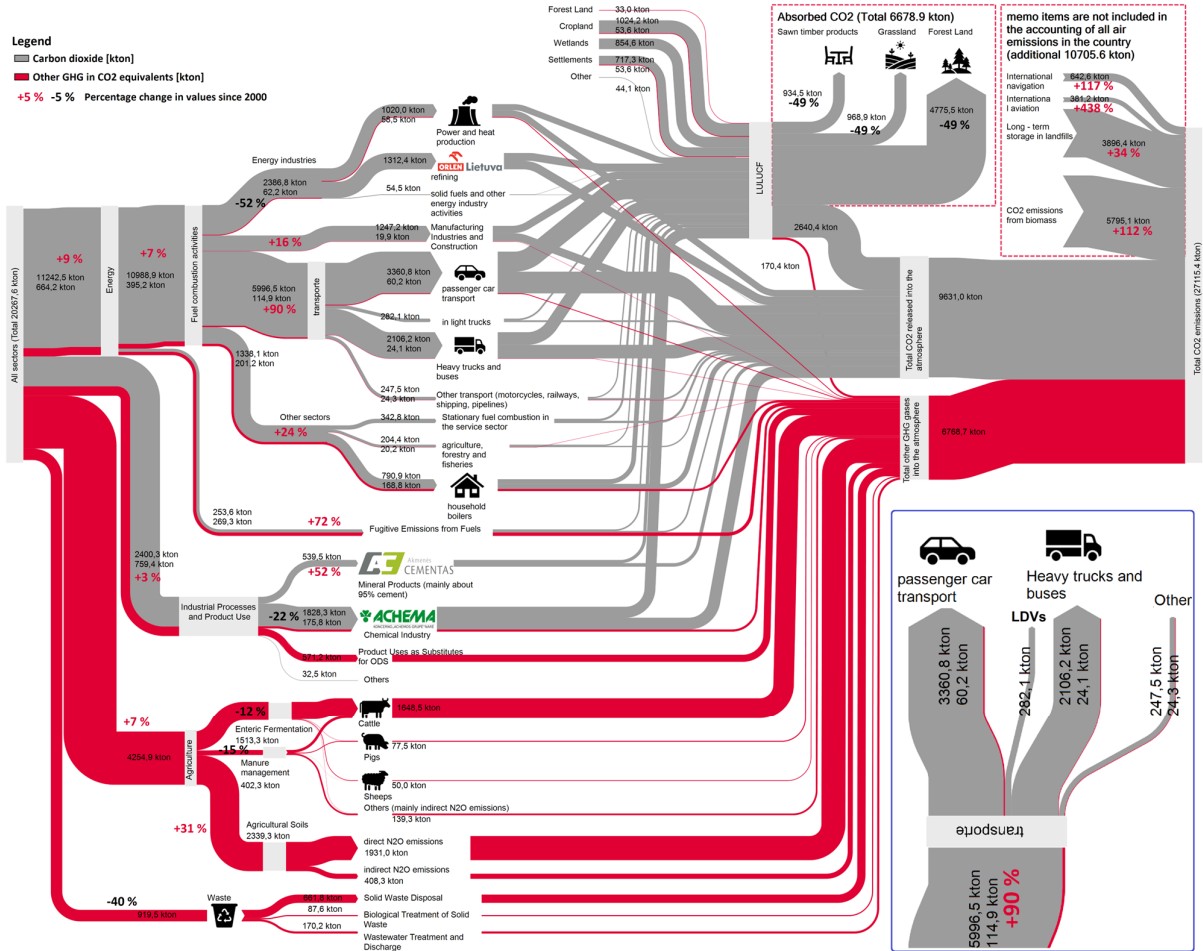

**Figure 5.** Lithuania's greenhouse gas balance [87].

Figure 5 suggests that the total $CO_2$ pollution from road transport almost reached 2789 kton in 2021 even if the bus fleet was not considered. $CO_2$ constitutes the dominant part of 99% of the greenhouse gases emitted via transport, while methane and $N_2O$ contribute only 0.25 kt and 0.175 kt, respectively [88]. The strategy for the development of Lithuanian transport and communications until 2050 [34] was to envision current and future challenges to prevent atmospheric pollution and to raise the question of a wider deployment of connected and automated transport.

Current scenario (2021). The publicly available data from the State Enterprise Regi-tra [89] show that the distribution of passenger cars in use in Lithuania, by fuel type, is as follows: 1,095,457 (68.1%) diesel cars, 370,757 (23.1%) petrol cars, 99,663 (6.2%) petrol cars equipped with an LPG, and 36,536 (2.3%) petrol HEVs. The battery-only cars account for only 0.3% (4841) of the total fleet of the LT car fleet despite the increased registration rates in recent years. The category N1 light-duty trucks with diesel engines accounts for 65,836, while the total stock of HDV categories N2 (35,451) and N3 (41,984) totals 77,435. Figure 6a

depicts that diesel fuel and gasoline are collectively responsible for 2,914.8 kton of $CO_2$ emissions (61.6% and 32.3%, respectively) from PCs. Diesel fuel is the biggest contributor to $N_2O$ pollution (23.09 t $CO_2$ Eq.) among the types of fuel evaluated. The Euro 4 and Euro 5 vehicles, which can be described as the most polluting (high emitter) compared to Euro 6 d temp, Euro 6d, and Euro 6 a/b/c, comprise the dominant amount of 70% of the PC fleet.

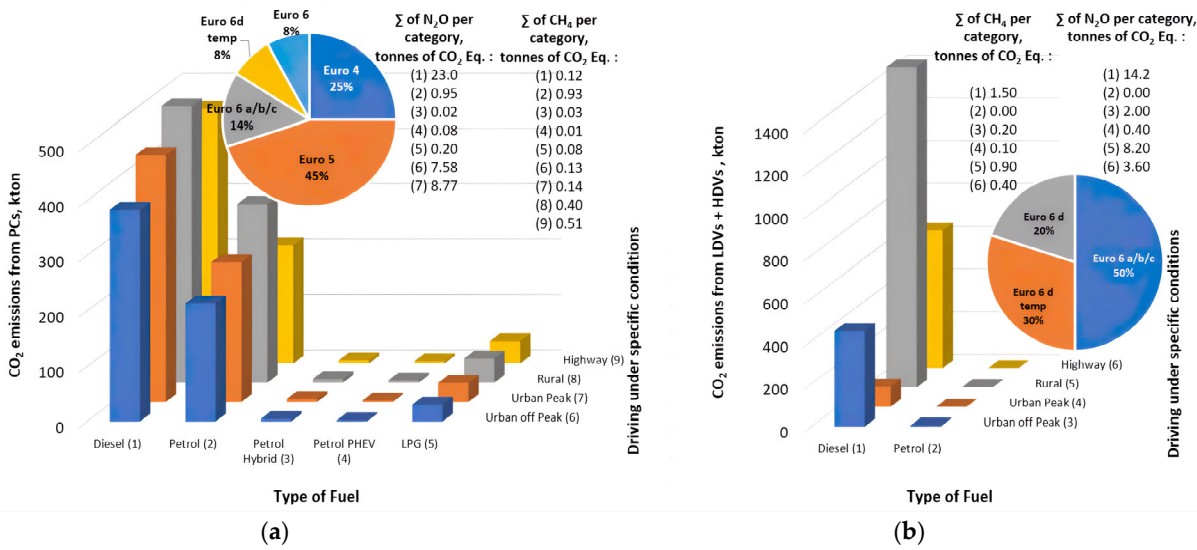

**Figure 6.** Total emission of GHG from road transport in 2021: (**a**) PC, (**b**) LDV and HDV.

Diesel fuel is the predominant type of fuel among LDV and HDV, as demonstrated in Figure 6b. The 'Rural' and 'Highway' driving can be described as significantly polluting modes that result in 2914 kton (80.42%) of $CO_2$, 24.25 t (82.8%) of $N_2O$, and 1.06 t (84.4%) of $CH_4$ emissions, with Euro 6 a/b/c being the category of vehicles with the highest pollution in 2021. The urban peak and rural driving conditions are responsible for 56.19% of all PC $CO_2$ emissions. These trends indicate the need not only to gradually renew the existing fleet of cars, which is on average 14 years old, but also to place special emphasis on the urgent need for a wider deployment of electric vehicles.

**Middle-term scenario (2030)**: This scenario is considered a medium-term option for Lithuania in the transition to sustainable fuels and transport. The passenger car fleet was assumed to decrease by approximately 18% (to 1,186,153 vehicles) compared to 2021 levels; 10% of the 2030 fleet will comprise battery-only vehicles, 50% petrol vehicles (~23% in 2021), 33.8% diesel vehicles (~68% in 2021), and 10% petrol–electric cars (~2% in 2021). The estimated emission levels from road transport will total 4,838 kton of $CO_2$, 35.3 tons of $CO_2$ Eq of $N_2O$, and 3.38 tons of $CO_2$ Eq of $CH_4$, depicted with respect to changing driving habits in Figure 7a,b. By 2030, a slow shift toward self-driving and shared PC use is expected [90].

The prediction of tailpipe $CO_2$ emissions assumes that, according to the realistic scenario for 2030 [34], urban and intercity (domestic) cargo services will experience 116% growth, followed by a 24% increase in international cargo capacity. The logistics sector will reach a point where it will have to expand the fleet from 77,435 vehicles in 2021 to 93,575 in 2030 to meet increasing demand or plan for growth. If the total number of heavy-duty diesel trucks of category N3 in Lithuania will increase by 30% as intended in this study, the annual $CO_2$ emissions from the most polluting category of vehicles may reach 3164 ktons (plus an additional 273 ktons from LDVs). Regarding the middle-term solutions, which can partially prevent growth in GHG emissions, the application of LNG as a fuel for dual-fuel engines seems to be the most realistic one. The 2030 scenario for heavy-duty transport attempts to reflect the possible outcomes if 50% of the LDV and HDV fleet is converted to run on dual fuel (best available technology not entailing excessive costs) and the remaining

50% are Euro 6d vehicles (N1, N2, N3) visualized in Figure 7b. The proposed scenario would allow one to reduce the $CO_2$ emission levels from LDVs by 35 ktons and to cut the carbon dioxide emission from HDVs by 395.5 ktons compared to the "diesel will remain dominant on the roads scenario". Regarding the goal to reach the $CO_2$ emission levels of 2021 (2517 kton) for HDVs, 80% of the fleet must be converted to run on dual fuel, which seems to be unrealistic in the 8-year time span. It is expected that there would still be a difference with a newer model year of diesel trucks (the fleet is assumed to be predominant by Euro 6d) when these newer diesel trucks, especially gas-powered trucks, would emit a few percent less $CO_2$, $N_2O$, and $CH_4$ due to technological improvements [91–94]. The prognosis that a stock share of 2 to 11% for zero-emission trucks will be available in Europe in 2030 may slightly adjust the overall emission levels for HDVs [95].

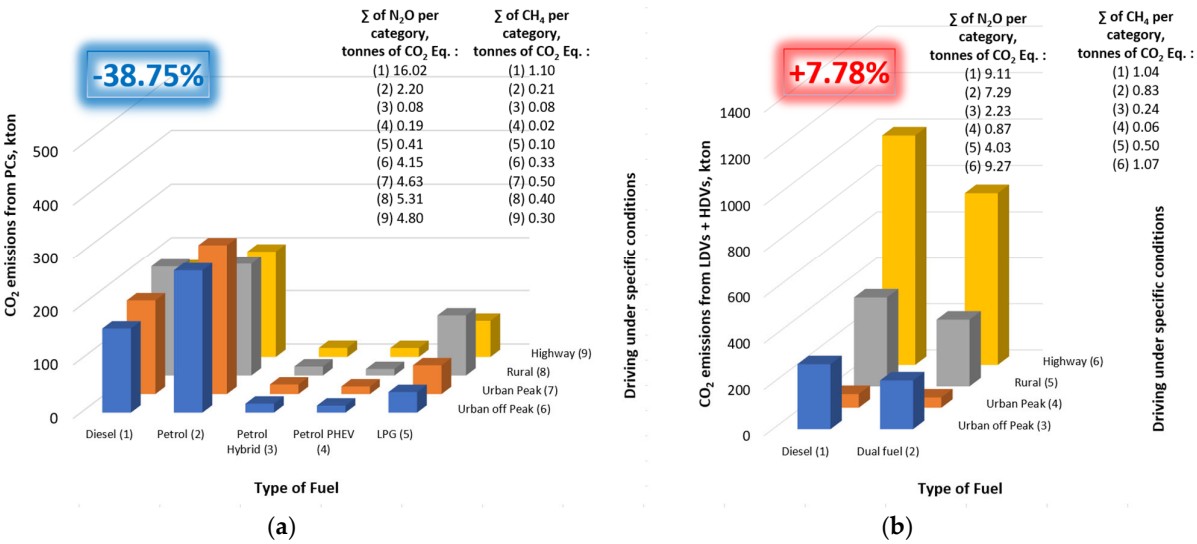

**Figure 7.** Total emission of GHG from road transport in 2030: (**a**) PC, (**b**) LDV and HDV.

**The long-term scenario (2050)** (see Figure 8) is characterized by uncertainty and complexity; however, it can help explore different alternative future pathways [96,97]. It is logical to assume that electric and fuel-cell electric vehicles, collectively called zero-emission vehicles (ZEVs), will play a dominant role in the sustainable low-carbon mobility of Lithuania in the middle of this century. Study [34] shows that the PC fleet can shrink by up to 45% by 2050 and will total 720–740,000. Despite previous predictions that new car sales in 2050 will have three broad categories (65.57% ZEV, 17.5% PHEV, and 16.93% ICEV) [21], the evaluation for Lithuania was carried out according to a plan approved by the European Union (29 June 2022) [98] to end the sale of vehicles with combustion engines (diesel and gasoline cars, as well as light commercial vehicles) by 2035 in Europe.

Given that a standard car lasts around 12 years or about 322,000 km, a span of 15 years seems to be reasonable enough to substitute the outdated internal combustion engine vehicles with 100% FCEVs (incl. AVs) and 100% BEVs (incl. AVs). The proposed substitutions can lead to achieving 100% carbon neutrality in the PC and LDV categories depicted in Figure 8.

The decarbonization of freight trucks still has many pathways to showcase different trajectories to reach the net zero target of 2050 [98]. It is estimated that the average rates of national road freight will increase by 161% followed by an increase of 81% in international freight capacity [34] compared to the 2020 levels, which, in turn, will demand on average 2.5–2.7 times higher numbers of LDVs and HDVs on Lithuanian roads. Most HDVs sold in 2050 are believed to be autonomous or human-powered BEVs or FCEVs, which will lead to a projected decrease in global emissions by 85–90% [21,62]. Other alternatives are switching to biofuels, ICEV-bio-LNG-D (dual fuel), HEV-D-ERS (hybrid catenary electric vehicles using electro fuels), and others [99]. Global HDV emissions are expected to decrease by

~6.8 times (to 423.4 kton of $CO_2$) from 2020 levels by 2050. The benefits of shifting to smaller vehicles on road freight are also possible [100].

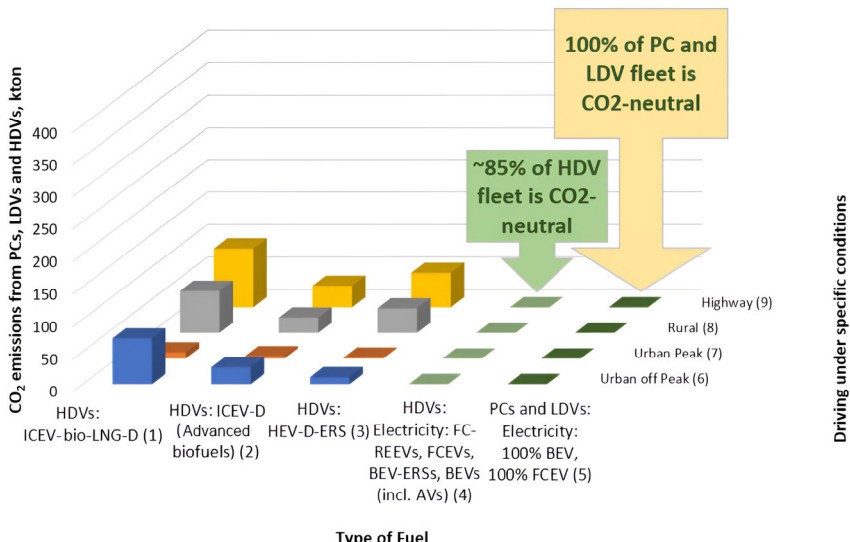

**Figure 8.** Total emission of GHG from road transport in 2050.

### 3.3. Lithuania's Standings in the Context of EU Countries

Lithuania falls within the top 30 countries in the world according to AV adoption readiness with the population size of 2.8 million [101]. The country is ranked 28th in the world and 17th among the EU member states. Regarding the four basic assessment criterions referred to in Table 4, the main strengths of Lithuania are the road quality (ranked 9th in the EU and 19th worldwide) and AV regulations (ranked 2nd—15th in the EU and 6th—21st worldwide).

Number 2 in the category 'Policy and legislation' means that the country has adopted the legislation/approval in place for testing AVs. Regarding the discussion in Section 3.2., a drastic increase by a factor of 10 compared to 2020 levels in public EV chargers is expected in the upcoming years, accounting for a total of 2143 EVCs per 1 million people (64 in 2021; 214 in 2022); refer to Table 4. The absence of AV companies with over USD 50 million investment is seen as the biggest challenge to increase the AV readiness score. Regarding the information presented in Section 3.3, the attraction of investments seems very realistic since Lithuania is a competitive fast-growing economy, ranking 39th worldwide in the Readiness for Frontier Technology Index and identified to have 'Smart Foreign Direct Investments' [102]. The overall performance position of the country in the category 'IMD World Digital Competitiveness' is being evaluated as 28th among 158 countries. Lithuania is placed within the first of four 25-percentile score groups, having a 'high' value of the index [17]. The 'Government AI Readiness Index' of 65.19 makes Lithuania the third country most ready in eastern Europe, with the regional average of 55.24 out of 100 compared to 47.42 for all countries [103]. The country ranks strongly in the 'Cybersecurity indicator' (11th out of all countries), supporting its strong regional ranking [103]. It is evident that numerous foreign direct investments focus on providing aid to accelerate R&D on AV; so, prior to the investment, the investing region is assessed for feasibility (qualified population, technology hub, resources, accessibility, trade network). The extensive feasibility analysis to shed focus on Lithuania, highlighting the noteworthy attributes in the context of a zero-emission fleet and autonomous mobility, serves novelty to this research that will gain attention from policy makers and investors and for research collaborations.

Our study expects that a well-established National Innovation System can generate several new technologies and services that are first tested and deployed in the domestic markets [10]. It confirms findings of other studies [104,105], concluding that the NIS

dynamics have different impacts on the achievement of sustainable development goals at different stages of economic development.

**Table 4.** The EU countries ranked according to their AV readiness, 2021 [101].

| Rank | Country | Policy and Legislation | Technology and Innovation | | | Consumer Acceptance | Infrastructure | | Score/10 |
| | | | Company Head Quarters | AV Patents | AV Companies with an over USD 50 Million Investment | | Infrastructure per Million of Population | Road Quality/10 | |
|---|---|---|---|---|---|---|---|---|---|
| 1 | France | 2 | 2 | 5499 | 2 | 5.9% | 679 | 5.4 | 7.37 |
| 2 | Germany | 4 | 1 | 13,817 | 1 | −2.3% | 537 | 5.3 | 6.74 |
| 3 | Sweden | 2 | 1 | 4693 | 1 | 0.0% | 1.006 | 5.3 | 6.34 |
| 4 | Austria | 2 | 1 | 693 | 1 | −2.3% | 923 | 6.0 | 5.85 |
| 5 | Finland | 1 | 1 | 1078 | 1 | 5.2% | 674 | 5.3 | 5.54 |
| 6 | The Netherlands | 2 | 0 | 3918 | 0 | −6.1% | 3,822 | 6.4 | 4.78 |
| 7 | Spain | 2 | 0 | 2352 | 0 | 18.9% | 173 | 5.7 | 4.47 |
| 8 | Ireland | 2 | 1 | 639 | 1 | 0.0% | 217 | 4.4 | 4.38 |
| 9–10 | Belgium | 2 | 0 | 1278 | 0 | 2.9% | 734 | 4.4 | 3.53 |
| 9–10 | Denmark | 2 | 0 | 1456 | 0 | −20.4% | 558 | 5.6 | 3.53 |
| 11 | Portugal | 2 | 0 | 237 | 0 | −19.7% | 240 | 6.0 | 3.08 |
| 12 | Poland | 2 | 0 | 161 | 0 | 13.3% | 45 | 4.3 | 2.5 |
| 13 | Italy | 1 | 0 | 2716 | 0 | −0.2% | 225 | 4.4 | 2.45 |
| 14−15 | Hungary | 2 | 0 | 81 | 0 | 9.0% | 133 | 4.0 | 2.32 |
| 14–15 | Greece | 2 | 0 | 120 | 0 | 4.8% | 31 | 4.6 | 2.32 |
| 16 | Estonia | 2 | 0 | 131 | 0 | −15.6% | 319 | 4.7 | 2.28 |
| 17 | Lithuania | 2 | 0 | 28 | 0 | 3.7% | 64 | 4.8 | 2.14 |
| 18–27 | Other countries: Slovakia, Luxembourg, Croatia, Republic of Cyprus, Slovenia, Czech Republic, Latvia, Bulgaria, Malta, Romania | | | | | | | | |

## 4. Conclusions

There are greater proximity results in the reduced consumption of road space and wider deployment of cleaner mobility, characterized by lower emissions and a more efficient use of resources for the second and subsequent vehicles in a platoon. The technology is already mature and needs development through trials, appropriate regulatory frameworks, and operating practices to enable the safe platooning of AVs on public roads. A systemic approach for the assessment of energy and transport sectors of Lithuania was proposed that gives new insights into innovative and economic performance of the country in terms of the exploration of the potential for autonomous mobility to reach the zero-emission fleets. The authors believe that the analysis of the development trajectory, correlations between key system variables, and the rate of change within the entire road transportation system can guide action toward sustainability. The initial statistical comparison of the status of Lithuania with the other leading regions of the world clearly shows that the country has the potential to become a contributor to the global reform of technology and innovation. The following conclusions can be drawn as a result of this research:

1.  This study on Lithuania's National Innovation System directs attention to the linkages or web of interaction within the overall innovation system. An understanding of NIS can help policy makers develop approaches for enhancing innovative performance in the knowledge-based economies of the future as it depends on the fluidity of knowledge flows—among the business environment, environment producing knowledge, and cooperation in science and technology.

2.  The expected transport fleet shift scenarios reveal that by 2030, there will be a significant reduction up to 38% in the $CO_2$ emission from PC but a raise of 7% in $CO_2$

emission from LDV and HDV. The slow transition toward the zero-emission fleet is expected to reflect evidently in 2050 with up to an ~85% reduction in $CO_2$ emission from HDV (compared to 2021 levels) and achievement of 100% carbon neutrality in the PC segment, which will be dominated mostly by zero-emission vehicles.

3. The energy efficiency targets reveal that ~20TWh of energy is expected to be saved by transportation reforms by 2030. The change in transportation will surge the energy demand by 7% in 2030, which will raise to 45% in 2050 in comparison to the energy consumption via transportation in 2020. In view of existing reforms and innovative entrepreneurship activities, which are considered a key factor of modern economic development, the Lithuanian energy sector is believed to be fully capable to ensure the projected energy demand for 2050.

4. Lithuania falls within the top 30 countries in the world according to AV adoption readiness with the population size of 2.8 million: according to road quality, it is ranked 9th in the EU and 19th worldwide, and according to AV regulation, it is ranked 2nd—15th in the EU and 6th—21st worldwide. Currently, the absence of AV companies with over USD 50 million investment is seen as the biggest challenge to increase the overall AV readiness score; however, our study expects that a well-established National Innovation System can generate several new technologies and services that are first tested and deployed in the domestic market, which will influence changes in foreign direct investment flows over the next decade.

**Author Contributions:** Conceptualization, L.R. and N.H.V.; methodology, L.R. and N.H.V.; software, N.H.V.; validation, L.R. and N.H.V.; formal analysis, L.R.; investigation, L.R. and N.H.V.; resources, L.R.; writing—original draft preparation, L.R. and N.H.V.; writing—review and editing, L.R. and N.H.V.; visualization, L.R. and N.H.V. All authors have read and agreed to the published version of the manuscript.

**Funding:** This research received no external funding.

**Institutional Review Board Statement:** Not applicable.

**Data Availability Statement:** The data are contained within the article.

**Acknowledgments:** Several impacting factors on regional level were initially collected during the implementation of The European Cooperation in Science and Technology (COST) project—CA16222 (WISE-ACT) Wider Impacts and Scenario Evaluation of Autonomous and Connected Transport (2017–2022, https://wise-act.eu/)—and later rearranged to fit this study. Many people have been involved in the various stages of this study, and we thank them all. The authors express thanks to Jurij Astafjev of UAB Teisingi Energetikos Sprendimai for providing data utilized in Figure 5 and for giving permission to publish it.

**Conflicts of Interest:** The authors declare no conflicts of interest.

## Abbreviations

**Latin symbols**

| | |
|---|---|
| AI | Artificial Intelligence |
| AM | Autonomous Mobility |
| AV | Autonomous Vehicle |
| $CH_4$ | Methane |
| $CO_2$ | Carbon Dioxide |
| $CO_2$ Eq | Carbon Dioxide Equivalent |
| $E_{HIGHWAY}$ | Emissions from Highway Driving Conditions |
| $E_{RURAL}$ | Emissions from Rural Driving Conditions |
| $E_{TOTAL}$ | Total Emissions from Different Driving Conditions |
| $E_{URBAN}$ | Emissions from Urban Driving Conditions |
| EU | European Union |
| Euro 4 to Euro 6 | European Emission Standards |
| $E_{HIGHWAY}$ | Emissions from Highway Driving Conditions |

| | |
|---|---|
| E$_{RURAL}$ | Emissions from Rural Driving Conditions |
| E$_{TOTAL}$ | Total Emissions from Different Driving Conditions |
| E$_{URBAN}$ | Emissions from Urban Driving Conditions |
| EU | European Union |
| Euro 4 to Euro 6 | European Emission Standards |
| EV | Electric Vehicle |
| EVC | Electric Vehicle Charging Station |
| FCEV | Fuel-Cell Electric Vehicle |
| GHG | Greenhouse Gas |
| HDV | Heavy-Duty Vehicle |
| HEV | Hybrid Electric Vehicle |
| ICEV | Internal Combustion Engine Vehicle |
| ICEV-bio-LNG-D | Dual-Fuel Vehicle (bio-LNG and Diesel Fuel) |
| ICT | Information and Communications Technology |
| LDV | Light-Duty Vehicle |
| LNG | Liquefied Natural Gas |
| M1, M2, M3 | Vehicle Categories Defined in the Consolidated Resolution on the Construction of Vehicles |
| N1, N2, N3 | Vehicle Categories Defined in the Consolidated Resolution on the Construction of Vehicles |
| N$_2$O | Nitrous Oxide |
| NIS | The National Innovation System |
| PC | Passenger Car |
| PHEV | Plug-in Hybrid Electric Vehicle |

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
