# Peer review of "A National Innovation System Concept-Based Analysis of Autonomous Vehicles’ Potential in Reaching Zero-Emission Fleets"

_technologies, doi:10.3390/technologies12020026_

Round 1

Reviewer 1 Report

Comments and Suggestions for Authors

Comments are attached. Thanks.

Reviewer 2 Report

Comments and Suggestions for Authors

The article is a detailed analysis of the potential for achieving zero fleet emissions of autonomous vehicles, based on the specifics of the National Innovation System in Lithuania. After a detailed study, it can be concluded that the article is prepared very carefully, the individual solved problems are logically related to each other. Although it is based on specific conditions in Lithuania, the article can be inspiring for similar studies in other countries or regions.

Therefore, I have no major comments regarding the substantive content of the article.

As far as formal requirements are concerned, I recommend considering whether the use of the abbreviation "NIS" in the title of the article is appropriate and immediately obvious to every reader. The meaning of this NIS abbreviation (National Innovation System) is obvious from the keywords and from another context, it could also be added to the abbreviations at the end of the article.

When writing references, e.g. line 32 [3-5], and others, according to the template, a hyphen [3–5] should be used instead of "-" as a separator.
